# In Situ Synthesis of C-N@NiFe_2_O_4_@MXene/Ni Nanocomposites for Efficient Electromagnetic Wave Absorption at an Ultralow Thickness Level

**DOI:** 10.3390/molecules28010233

**Published:** 2022-12-27

**Authors:** Qing Sun, Xin Yang, Tie Shu, Xianfeng Yang, Min Qiao, Dashuang Wang, Zhaohui Liu, Xinghua Li, Jinsong Rao, Yuxin Zhang, Pingan Yang, Kexin Yao

**Affiliations:** 1Multi-Scale Porous Materials Center, Institute of Advanced Interdisciplinary Studies, School of Chemistry and Chemical Engineering, Chongqing University, Chongqing 400044, China; 2State Key Laboratory of Photon-Technology in Western China Energy, School of Physics, Northwest University, Xi’an 710127, China; 3College of Material Science and Engineering, Chongqing University, Chongqing 400044, China; 4School of Automation, Chongqing University of Posts and Telecommunications, Chongqing 400065, China

**Keywords:** electromagnetic wave absorption, MXene, NiFe_2_O_4_

## Abstract

Recently, the development of composite materials composed of magnetic materials and MXene has attracted significant attention. However, the thickness and microwave absorption performance of the composite is still barely satisfactory. In this work, the C-N@NiFe_2_O_4_@MXene/Ni nanocomposites were successfully synthesized in situ by hydrothermal and calcination methods. Benefiting from the introduction of the carbon-nitrogen(C-N) network structure, the overall dielectric properties are improved effectively, consequently reducing the thickness of the composite while maintaining excellent absorption performance. As a result, the minimum reflection loss of C-N@NiFe_2_O_4_@MXene/Ni can reach −50.51 dB at 17.3 GHz at an ultralow thickness of 1.5 mm, with an effective absorption bandwidth of 4.95 GHz (13.02–18 GHz). This research provides a novel strategy for materials to maintain good absorption performance at an ultralow thickness level.

## 1. Introduction

With the development of telecommunication and microelectronic technologies, the problems of electromagnetic interference and radiation gradually become more serious. Excess electromagnetic waves can not only disrupt the operation of other electronic devices, but also cause harm to human health [1,2,3,4]. In order to reduce the negative effects of electromagnetic waves, microwave absorption materials (MAMs) are applied widely. MAMs can absorb microwave and electromagnetic energy. In general, the ideal MAMs should have the characteristics of light weight, thin thickness, strong absorption, and wide frequency band [5,6,7].

Two-dimensional transition metal carbides (MXene), as new emerging layered material, are widely used in various fields such as electrochemical energy storage [8], electromagnetic shielding [9], biomedical [10], etc. MXene is produced by etching and stripping its precursor MAX, where M is a transition metal element, A is a main group element, usually Al or Si, and X denotes C or N [11]. The MX layer has a strong bonding and is in a stable chemical state. The A-layer group, as a reactive site, is susceptible to selective etching by hydrofluoric acid, hydrochloric acid/lithium fluoride, molten salt, etc., which breaks its chemical bonds with M and X layers. The MAX layer spacing is significantly increased after etching, and the MXene sheet layer is obtained. Because of its outstanding electrical conductivity, abundant functional groups, and high polarization anisotropy [12], MXene can be used to manufacture microwave absorbing materials. However, a single loss mechanism limits its performance.

At present, many investigations have proved that combining dielectric materials (MXene, graphene, and SiC) with magnetic materials (Ni [13], Fe_2_O_3_ [14], Fe_3_O_4_ [15], and NiFe_2_O_4_ [16,17]) is an effective way to improve absorption performance. The synergistic effect of the composites not only enhances magnetic and dielectric losses, but also improves impedance matching [18]. For instance, Zhang et al. [19] prepared Fe_3_O_4_@Ti_3_C_2_T_x_ nano-composites by the solvothermal process. The prepared composite has a minimum reflection loss (*R_L_*_min_) value of −57.2 dB(15.7 GHz) and an effective absorption bandwidth of 1.4 GHz at a thickness of 4.2 mm. Sandwich-like CoFe@Ti_3_C_2_T_x_ composite was successfully synthesized by in situ reduction by Zhou et al. [20], the *R_L_*_min_ value of −36.29 dB at 8.56 GHz could be achieved with a thickness of 2.2 mm. Xiao et al. [21] fabricated MXene-CNTs/Ni hybrid, which had an *R_L_*_min_ performance of −56.4 dB at 2.4 mm thickness. Although the composites formed by combining magnetic materials with MXene have good absorption properties, it is a great challenge for the composites to maintain excellent absorption performance at an ultralow thickness.

In order to obtain MAMs at ultralow thickness level, the factors that affect the thickness have been discussed. According to the theory of transmission lines, the following equation is obtained [22,23].
(1)$$Z_{in}=Z_{0}\sqrt{\frac{\mu_{r}}{\varepsilon_{r}}}\tanh\left(\dot{J}\frac{2\pi h}{\lambda }\sqrt{\mu_{r}\varepsilon_{r}}\right)$$
(2)$$R_{L}\left(dB\right)=20\log_{10}\left(\left|\frac{Z_{in}-Z_{0}}{Z_{in}+Z_{0}}\right|\right)$$
where $Z_{0}$ is free space wave impedance; $Z_{in}$ is the input impedance; $\varepsilon_{r}$ represents the relative complex permittivity; $\mu_{r}$ represents the complex permeability; h represents the thickness of absorber; λ is the wavelength of electromagnetic wave and $R_{L}$ is the reflectivity. According to the above two formulas, the conditions of general matching absorption are discussed. In an atmospheric environment, the dielectric constant of magnetic material is greater than magnetic conductivity. When $\left|\varepsilon \right|>\left|u\right|$, the optimal absorption thickness is as follows:(3)λh=kRe⁡εu−c
(4)$$R_{e}\sqrt{\varepsilon \mu }=\sqrt{\frac{\varepsilon^{\prime }\mu^{\prime }}{2}}\sqrt{\left(1-tg\delta_{s}tg\delta_{\mu }\right)+\sqrt{1+tg^{2}\delta_{s}tg^{2}\delta_{\mu }+tg^{2}\delta_{s}+tg^{2}\delta_{\mu }}}$$
where k and c are constant. The results show the critical factors of optimal thickness are the product of the real part of the complex permittivity of the material and the real part of the complex permeability of the material ($\varepsilon^{\prime }\mu^{\prime }$) and the ratio of dielectric loss to magnetic loss (tanδε/tanδμ). When the value of $\varepsilon^{\prime }\mu^{\prime }$ is big and tanδε/tanδμ is greater than or less than 1, the minimum thickness can be obtained.

Accordingly, in this work, the C-N network structure is employed to change the overall dielectric properties of the MAMs. C-N@NiFe_2_O_4_@MXene/Ni was prepared in situ by hydrothermal and calcination methods. Due to the adjustment of dielectric properties, the value of $\varepsilon^{\prime }\mu^{\prime }$ increases and tanδε/tanδμ further away from 1 to achieve optimal absorption thickness value. In addition, the introduction of C-N network structure is conducive for enhancing dielectric loss. The multicomponent three-dimensional structure composed of C-N network structure, NiFe_2_O_4_ and MXene/Ni can generate dipole polarization and multiple scattering [24]. The result verifies that the synergistic effect of C-N network structure, NiFe_2_O_4_ and MXene/Ni can effectively improve the electromagnetic wave absorption performance and reduce its the thickness.

## 2. Results Discussion

The X-ray diffraction (XRD) patterns of NiFe LDH@MXene/Ni and C-N polymer@NiFe LDH@MXene/Ni are shown in Figure 1a. The characteristic diffraction peak of (002) indicates the successful preparation of MXene [25]. MXene was prepared by using Lewis acidic etching route. When immersing Ti_3_AlC_2_ MAX precursor in molten NiCl_2_ at 750 °C, the Al atoms weakly bonded to Ti in the Ti_3_AlC_2_ are oxidized into Al^3+^ cations by the Lewis acid Ni^2+^, resulting in the formation of AlCl_3_ phase and concomitant reduction of Ni^2+^ to nickel metal [26]. After reacting with dilute hydrochloric acid, the great mass of nickel on MXene was removed. A small amount of nickel that grows in the crystal lattice still remained. The diffraction peaks at 44.3°, 51.6°, and 76.1° belong to the (111), (200), and (220) crystalline planes of Ni (PDF#89-7128), respectively [22]. Obviously, nickel ions have been successfully converted into nickel metal. The diffraction peaks at 11.4°, 22.9°, 34.5°, and 59.9° belong to the (003), (006), (012), and (113) crystalline planes of NiFe LDH (PDF#40-0215), respectively, which illustrates the successful synthesis of NiFe LDH based on MXene/Ni substrate as expected. From NiFe LDH@MXene/Ni to C-N polymer@NiFe LDH@MXene/Ni, the diffraction peak of (003) crystalline plane shifts to a small angle (Appendix A), indicating that the introduction of C-N polymer makes the interlayer spacing of NiFe LDH increase. Furthermore, as shown in Figure 1b, the NiFe LDH has been successfully transformed into NiFe_2_O_4_ after thermal treatment in a nitrogen atmosphere. The diffraction peaks at 35.7°, 43.6°, and 62.9° belong to the (311), (400), and (440) crystalline planes of NiFe_2_O_4_ (PDF#10-0325) [27]. Notably, the XRD plot of C-N@NM has an additional carbon peak (PDF#12-0212), obtained from the C-N polymer after thermal treatment at a high temperature. In addition, X-ray photoelectron spectroscopy (XPS) was used to characterize the surface elemental composition and chemical state of the experimental samples. As shown in Appendix A, the XPS survey spectra of both NM and C-N@NM exhibit the existence of Ni, Fe, Ti, C, and O elements. Compared with the NM, the wide-scan XPS spectrum of the C-N@NM has an extra peak (N 1s). The high-resolution XPS spectra of C 1s of C-N@NM (Figure 1c) can be deconvoluted into three peaks. The binding energies at 284.8 eV, 285.7 eV and 287.6 eV are assignable to the graphitized C-C/C=C, C-O and C=O, respectively [28]. For the high-resolution XPS spectra of N 1s (Figure 1d), the peaks at 398.3 eV and 399.7 eV are assigned to sp^3^ C-N and sp^2^ C-N. During the formation of C-N@NM, nitrogen atoms replace the carbon atoms in sp^2^ and sp^3^ C-C bonds, and consequently, the prepared C-N network contains sp^2^ and sp^3^ C-N bonds [29,30]. In the spectra of Ni 2p (Figure 1e), the binding energies at 854.6 and 871.1 eV are assigned to Ni 2p_3/2_ and Ni 2p_1/2_. The divalent (Ni^2+^ 2p_3/2_ and Ni^2+^ 2p_1/2_) oxidative states are at 856.4 eV and 874.8 eV peaks. Moreover, the binding energies located at 860.6 and 878.5 eV are ascribed to shake-up satellite peaks (marked as “Sat.”) [31,32]. The spectrum of Fe 2p is exhibited in Figure 1f. The two large peaks at 710.7 and 724.5 eV are ascribed to Fe 2p_3/2_ and Fe 2p_1/2_, respectively. Furthermore, the peaks of Fe 2p_3/2_ and Fe 2p_1/2_ can be deconvoluted into two peaks, both corresponding to Fe^3+^. Meanwhile, the peaks at 719.2 and 732.6 eV are satellite peaks.

Additionally, the as-synthesized materials were further characterized by scanning electron microscopy (SEM). As shown in Figure 2a, the Ti_3_AlC_2_ MAX raw material features an aggregated block. After Lewis acidic etching, the molten NiCl_2_ removed the Al atoms of MAX. The raw material obviously changes to layered MXene/Ni (Figure 2b). Appendix A and Figure 2c,d are the NiFe LDH@MXene/Ni and C-N polymer@NiFe LDH@MXene/Ni, respectively. The morphology of NiFe LDH growing on MXene/Ni is lamellar. The growth mechanism of NiFe LDH is as follows. With the increase of reaction temperature, urea gradually hydrolyzes to OH^−^ and CO_3_^2−^ (CO(NH_2_)_2_ + H_2_O → 2NH_3_ + CO_2_, NH_3_·H_2_O → NH_4_^+^ + OH^−^, CO_2_ + H_2_O → CO_3_^2−^ + H^+^). The Ni^2+^ and Fe^3+^ react preferentially with OH^−^ to form monomeric nickel-iron hydroxide. Meanwhile, the monomers nucleate and aggregate on the surface of MXene/Ni. As the heating time increases, the hydrolysis of urea provides more OH^−^ ions, which promotes the continued growth of nickel-iron hydroxide particles. Eventually, nickel-iron hydroxide gradually formed NiFe-LDH nanosheets. It should be noted that the reason why NiFe-LDH can grow on MXene/Ni is that MXene has abundant surface groups (-OH, -O and/or -F, etc.) [33], that have hydroxyl linkage with NiFe-LDH. For C-N polymer@NiFe LDH@MXene/Ni, after calcination at 900 °C in N_2_, the NiFe-LDH nanosheets were in situ converted to spherical NiFe_2_O_4_ particles (Appendix A and Figure 2e,f), and the C-N polymer simultaneously was calcined to C-N network structure. Compared with NM (Appendix A), the NiFe_2_O_4_ particles of C-N@NM are denser and more homogeneous. This is because the C-N network structure protects NiFe-LDH from excessive collapse during calcination. It can be seen that C-N polymer is not only a precursor of the C-N network structure, but also has a positive modulating effect on the dispersion of NiFe_2_O_4_ nanoparticles.

In order to further reveal microstructure of as-prepared C-N@NiFe_2_O_4_@MXene/Ni, the transmission electron microscopy (TEM) was employed. The TEM image of C-N@NM (Figure 3a) again clearly demonstrates that C-N@NM consists of three parts: C-N network structure, NiFe_2_O_4_, and MXene/Ni. The C-N network structure tends to be an amorphous structure, contributing to dielectric and conductivity losses. Meanwhile, the high-resolution TEM image of C-N@NM is shown in Figure 3b. The 0.28 nm lattice fringe matches the (220) lattice plane of NiFe_2_O_4_. The lattice fringe of 0.36 nm belongs to MXene. Furthermore, the scanning TEM and corresponding elemental mapping images (Figure 3c) further indicate the distribution of C, N, Fe, Ni and Ti. It can be seen that the Ni and Fe elements are distributed in the interior of the sphere structure. Meanwhile, C is wrapped around the outside. Combining the result of TEM with the above analysis of XRD and SEM, it can be determined that the NiFe_2_O_4_ spheres are grown with MXene/Ni as the substrate and wrapped by C-N network structure. The C-N@NiFe_2_O_4_@MXene/Ni has a three-dimensional interconnected multi-interface structure, which facilitates multiple interfacial polarizations.

To investigate the electromagnetic wave absorbing performance of NM and C-N@NM, the *R_L_* was evaluated. The calculation formulas are as follow [34,35]:(5)$$Z_{in}=Z_{0}\sqrt{\frac{\mu_{r}}{\varepsilon_{r}}}\tanh\left(\dot{J}\frac{2\pi h}{\lambda }\sqrt{\mu_{r}\varepsilon_{r}}\right)$$
(6)$$Z=\left|\frac{Z_{in}}{Z_{0}}\right|=\sqrt{\left|\mu_{r}/\varepsilon_{r}\right|}\tan h\left[j\left(\frac{2\pi fd}{c}\right)\sqrt{\mu_{r}\varepsilon_{r}}\right]$$
(7)$$R_{L}\left(dB\right)=20\log_{10}\left(\left|\frac{Z_{in}-Z_{0}}{Z_{in}+Z_{0}}\right|\right)$$
where d represents the thickness of absorber; f represents the frequency of the microwave; c is the velocity of light. Figure 4b, d separately show the *R_L_* of NM and C-N@NM with a thickness of 1–5 mm in the range of 2–18 GHz and its corresponding three-dimensional projection diagrams (Figure 4a,c). According to the previous literature [36], if the *R_L_* value is less than −10 dB, 90% of incident electromagnetic waves can be completely absorbed, the corresponding frequency width is called the effective absorption bandwidth (EAB). Based on the criterion, the absorbing properties of NM and C-N@NM are discussed. For NM (Figure 4b), when the thickness is 5 mm, the *R_L_*_min_ value is −25.78 dB with the EAB of 4.87 GHz (3.59–8.46 GHz). Surprisingly, C-N@NM (Figure 4d), at an ultra-thin level of 1.5 mm, exhibits dramatically microwave absorbing performance. The *R_L_*_min_ value can reach −50.51 dB at 17.3 GHz, and the EAB is slightly increased to 4.95 GHz (13.02–18 GHz), which is superior to that of NM, indicating that the multi-component C-N@NM has better electromagnetic wave absorbing performance at an ultralow thickness level due to the introduction of the C-N network structure.

For further determining the microwave absorption performance, there are two important parameters, which are relatively complex permittivity ($\varepsilon_{r}=\varepsilon^{\prime }-j\varepsilon^{\prime \prime }$) and relative complex permeability ($\mu_{r}=\mu^{\prime }-j\mu^{\prime \prime }$). The real part ($\varepsilon^{\prime }$ and $\mu^{\prime }$) is related to the ability to store electromagnetic wave energy, and the imaginary part ($\varepsilon^{\prime \prime }$ and $\mu^{\prime \prime }$) represents the ability of consuming the electromagnetic wave energy [37,38]. As can be seen from Figure 5a,b, the $\varepsilon^{\prime }$ and $\varepsilon^{\prime \prime }$ values of C-N@NM are significantly larger than those of NM, indicating that C-N@NM has stronger energy storage capacity and dielectric loss capability. In addition, the polarization peaks of C-N@NM appear in Figure 5b, demonstrating the existence of multiple polarization relaxation and conductance loss in C-N@NM. Figure 5c,d show that both $\mu^{\prime }$ and $\mu^{\prime \prime }$ of C-N@NM are smaller than NM, indicating that the magnetic loss is reduced from NM to C-N@NM. In general, compared with NM, although the introduction of C-N conductive network slightly reduces the magnetic loss of C-N@NM, it greatly enhances the dielectric constant of the material, thereby increasing the dielectric loss and resulting in enhanced overall absorbing performance. Additionally, the degrees of dielectric and magnetic dissipation were separately measured by dielectric loss tangent (tan*δε* = *ε*″/*ε*′) and magnetic loss tangent (tan*δμ* = *μ*″/*μ*′) [39,40,41]. Figure 5e shows that the tan*δε* value of C-N@NM is greater than that of NM. It can be seen from Figure 5f that the tan*δμ* value of C-N@NM is smaller than that of NM. The tanδμ/tanδμ ratio of C-N@NM is further away from 1 in the 2–18 GHz range (Figure 5g), proving that dielectric loss plays a dominant role in microwave losses. Meanwhile, the $\varepsilon^{\prime }\mu^{\prime }$ value of C-N@NM is significantly greater than NM (Figure 5h). The result indicates that C-N@NM possesses perfect impedance matching conditions at an optimal thickness value. Thus, the *R_L_* value of C-N@NM can reach −50.51 dB when the thickness is just 1.5 mm.

According to the Debye theory, the Cole-Cole curve (*ε*″-*ε*′) can be used to study the mechanism of dielectric loss. The relationship between *ε*′ and *ε*″ is shown in the following equation [42]:(8)$${\left(\varepsilon^{\prime }-\frac{\varepsilon_{s}+\varepsilon_{\infty }}{2}\right)}^{2}+={\left(\frac{\varepsilon_{s}-\varepsilon_{\infty }}{2}\right)}^{2}$$
where *ε*_s_ and *ε*_∞_ are the static permittivity and relative permittivity in higher frequency regions, respectively. The Cole–Cole curves of NM and C-N@NM are exhibited in Figure 6a,b, which both have multiple semicircles in the curves. Each semicircle represents a Debye dipolar relaxation process [43]. Moreover, it is clear that each Cole–Cole curve consists of semicircles and straight tails, indicating that in addition to Debye relaxation, other dielectric loss mechanisms also make contributions to dielectric loss, such as dipole polarization and electron polarization [44]. Potential differences exist at multiple interfaces of the C-N network structure, NiFe_2_O_4_ and MXene/Ni, leading to the accumulation of free electrons at the interfaces, and causing additional polarization loss.

The attenuation constant (α) is a crucial parameter used to evaluate the absorption capacity of a material. The calculation equation is as follows [45]:(9)$$\alpha =\frac{\sqrt{2}\pi f}{c}\sqrt{\mu^{\prime \prime }\varepsilon^{\prime \prime }-\mu^{\prime }\varepsilon^{\prime }+\sqrt{{\left(u^{\prime \prime }\varepsilon^{\prime \prime }-u^{\prime }\varepsilon^{\prime }\right)}^{2}+{\left(u^{\prime }\varepsilon^{\prime \prime }+u^{\prime \prime }\varepsilon^{\prime }\right)}^{2}}}$$

If the attenuation constant is larger, the ability to dissipate electromagnetic waves will be stronger. The attenuation constant of NM and C-N@NM are displayed in Figure 6c. Although in the 2–7.5 GHz range, the NM has a slightly higher attenuation constant than C-N@NM, the attenuation constant of C-N@NM is significantly greater than that of NM in the range of 7.5–18 GHz, indicating that the introduction of C-N network structure is beneficial to improve the absorption performance of the material in high frequency band. Additionally, for C-N@NM, the fluctuation of the attenuation constant in the high frequency range indicates that the polarization relaxation plays a good role in the attenuation process of electromagnetic waves at a specific frequency. The presence of multiphase interfaces enhances interface polarization and dipole polarization so that electromagnetic waves have better conduction losses as they pass through more interfaces.

On the basis of the above discussion, the reasonable microwave absorption mechanism of C-N@NM material is described. Firstly, the introduction of C-N conductive network structure not only contributes to enhancing the conductive loss, but also can effectively improve and optimize the impedance matching of the material, thus achieving higher microwave absorption performance at an optimal thickness. Secondly, the three-dimensional interconnected multi-interface structure can cause the multi-interface reflection of electromagnetic waves. The multiple reflections of electromagnetic waves will increase the transmission distance, resulting in more energy dissipation. Meanwhile, due to the existence of multiple components, amounts of heterogeneous interfaces can generate interfacial polarization, which is essential to wave absorption. Thirdly, the synergistic effect of C-N network structure, NiFe_2_O_4_ and MXene/Ni contributes to the attenuation ability of the material to electromagnetic waves. With the combined effect of the above factors, the C-N@NM can exhibit excellent wave absorption performance at an ultralow thickness level.

## 3. Materials and Methods

### 3.1. Materials

The materials required, namely Fe(NO_3_)_3_·9H_2_O, Ni(NO_3_)_2_·6H_2_O, NiCl_2_, NH_4_F, urea, p-toluenesulfonic acid(PA), and metanilic acid(MA), were all obtained from Shanghai Aladdin Biochemical Technology Co., Ltd. (Shanghai, China). Ti_3_AlC_2_ (MAX) was obtained from 11 Technology Co., Ltd. (Changchun, China). All reagents were analytical grade and used without any further purification.

### 3.2. Synthesis of MXene

MXene was synthesized by using a molten salt etching method [26]. 1 g of Ti_3_AlC_2_ MAX and 2.1 g of NiCl_2_ were mixed and ground for 10 min. Then, 0.76 g of KCl and 0.6 g of NaCl were added to the above mixture. The mixture was continued to be ground for 20 min to make it homogeneous. In an argon atmosphere, the powder mixture was heated at 750 °C for 24 h to obtain MXene/Ni. The MXene/Ni was then washed by 0.1 M HCl solution to remove the residual Ni particles. The resulting solution was filtered with a microfiltration membrane. Finally, the powder was dried under vacuum at room temperature for 24 h.

### 3.3. Synthesis of NiFe LDH@MXene/Ni and C-N polymer@NiFe LDH@MXene/Ni

The materials were synthesized in an autoclave by a solvothermal route. 0.116 g of Ni(NO_3_)_2_·6H_2_O, 0.116 g of Fe(NO_3_)_2_·9H_2_O, 0.48 g of urea, 0.13 g of NH_4_F, and 0.2 g MXene/Ni were magnetically stirred in 60 mL of deionized water for 1 h to obtain a homogeneous solution A. The solution A was transferred to a 100 mL Teflon-lined stainless-steel autoclave and heated at 120 °C for 3 h. Subsequently, the resulting precipitate was collected by centrifugation, washed with deionized water for several times, and dried at 60 °C for 6 h to obtain NiFe LDH@MXene/Ni. For producing C-N polymer@NiFe LDH@MXene/Ni, the additional 50 mmol of PA and 50 mmol of MA were slowly added to the above solution A and stirred constantly to obtain a homogeneous solution B. The pH of solution B was adjusted to 7.0. The neutral solution was then transferred to Teflon-lined stainless-steel autoclave, and heated 3 h at 120 °C. The precipitate obtained by centrifugation was washed and dried to receive C-N polymer@NiFe LDH@MXene/Ni.

### 3.4. Synthesis of NiFe_2_O_4_@MXene/Ni and C-N@NiFe_2_O_4_@MXene/Ni

The obtained NiFe LDH@MXene/Ni and C-N polymer@NiFe LDH@MXene/Ni powders were separately put into a tubular furnace, raised to 900 °C with a heating rate (5 °C min^−1^) and kept for 2 h in nitrogen. The NiFe_2_O_4_@MXene/Ni and C-N@NiFe_2_O_4_@MXene/Ni calcined powders were collected and marked as NM and C-N@NM, respectively.

### 3.5. Characterization

The microscopic morphology and particle size of the samples were characterized using scanning electron microscopy (SEM, Helios 5 CX, Thermo Scientific, Waltham, MA, USA) and transmission electron microscopy (TEM, Talos, F200S, Thermo Scientific, Waltham, MA, USA G2). The crystal structure and phase composition were analyzed using X-ray diffraction (XRD, Empyrean, Panalytical B.V., Almelo, The Netherlands) with Cu Kα radiation between 5° and 80° (40 kV; 40 mA; 5° min^−1^). Additionally, X-ray photoelectron spectroscopy (XPS, K-Alpha, Thermo Scientific, Waltham, MA, USA) was obtained using a Thermo Scientific Kα energy spectrometer paired with an X-ray source of monochromatic Al-K_α_. The NM (50 wt%) or C-N@NM (50 wt%) were uniformly mixed with paraffin wax (50 wt%) and pressed into a concentric ring with an outer diameter of 7.0 mm, and an inner diameter of 3.04 mm. The electromagnetism parameters of NM and C-N@NM, in the frequency range of 2–18 GHz, were obtained using a vector network analyzer (Agilent N5234A, Santa Clara, CA, USA) using the coaxial method.

## 4. Conclusions

The three-dimensional interconnected multi-interface C-N@NM structure was successfully synthesized in situ by hydrothermal and calcination methods. Compared with NM, the introduction of C-N network structure not only changes the overall dielectric properties of the material, thereby reducing the thickness of the material, but also facilitates interfacial polarization. This is the reason why the C-N@NM achieves excellent absorption performance at an ultralow thickness level. The *R_L_*_min_ of C-N@NM absorber reaches −50.51 dB at the thickness of only 1.5 mm. The proposed strategy in this work can provide a guidance and possibility for the preparation of absorbers with low thickness and highly efficient electromagnetic wave absorption performance in the future.

## Figures and Tables

**Figure 1 molecules-28-00233-f001:**
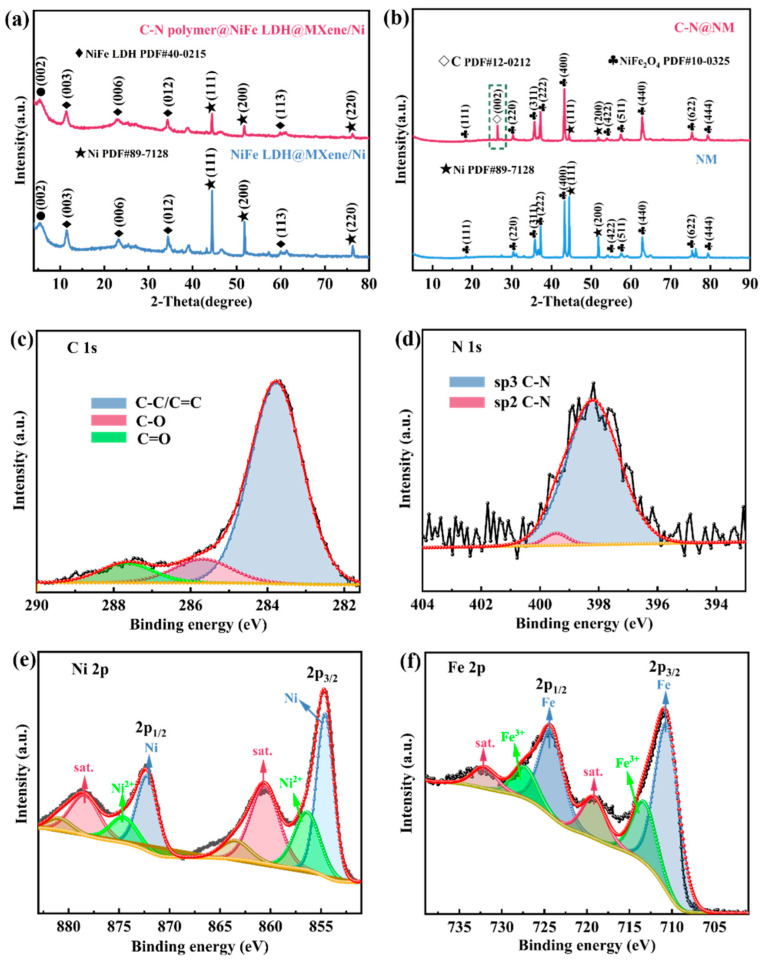
XRD patterns: (**a**) NiFe LDH@MXene/Ni and C-N polymer@NiFe LDH@MXene/Ni, (**b**) NM and C-N@NM; XPS spectra of C-N@NM: (**c**) C 1s; (**d**) N 1s; (**e**) Ni 2p; (**f**) Fe 2p.

**Figure 2 molecules-28-00233-f002:**
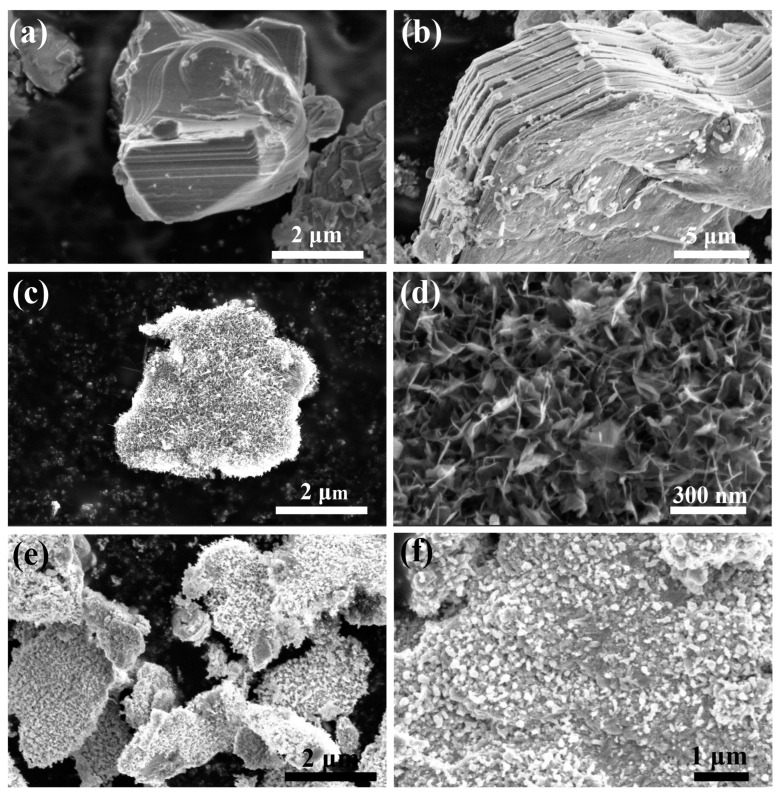
SEM images of (**a**) Ti_3_AlC_2_ MAX, (**b**) MXene/ Ni; (**c**,**d**) C-N polymer@NiFe LDH@MXene/Ni; (**e**,**f**) C-N@NM.

**Figure 3 molecules-28-00233-f003:**
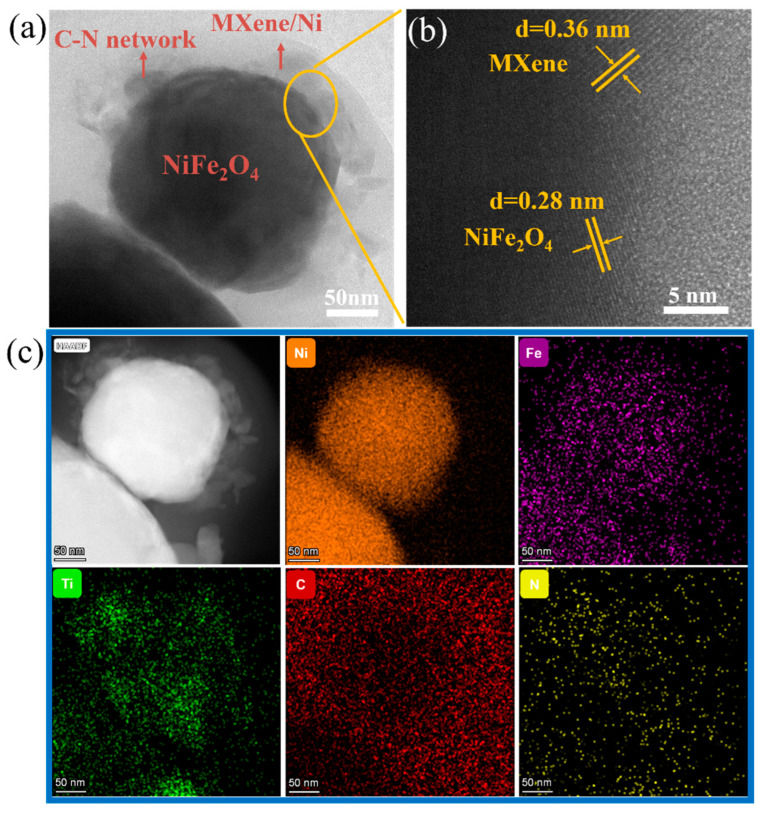
(**a**) TEM image of C-N@NM; (**b**) HRTEM image of C-N@NM; (**c**) STEM and corresponding elemental mapping images of C-N@NM.

**Figure 4 molecules-28-00233-f004:**
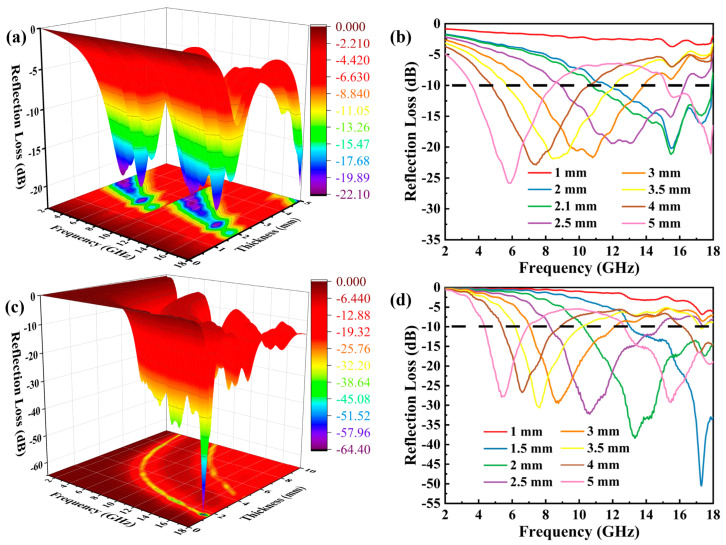
The 3D reflection loss pattern of (**a**) NM; (**c**) C-N@NM; Reflection loss curves of (**b**) NM; (**d**) C-N@NM.

**Figure 5 molecules-28-00233-f005:**
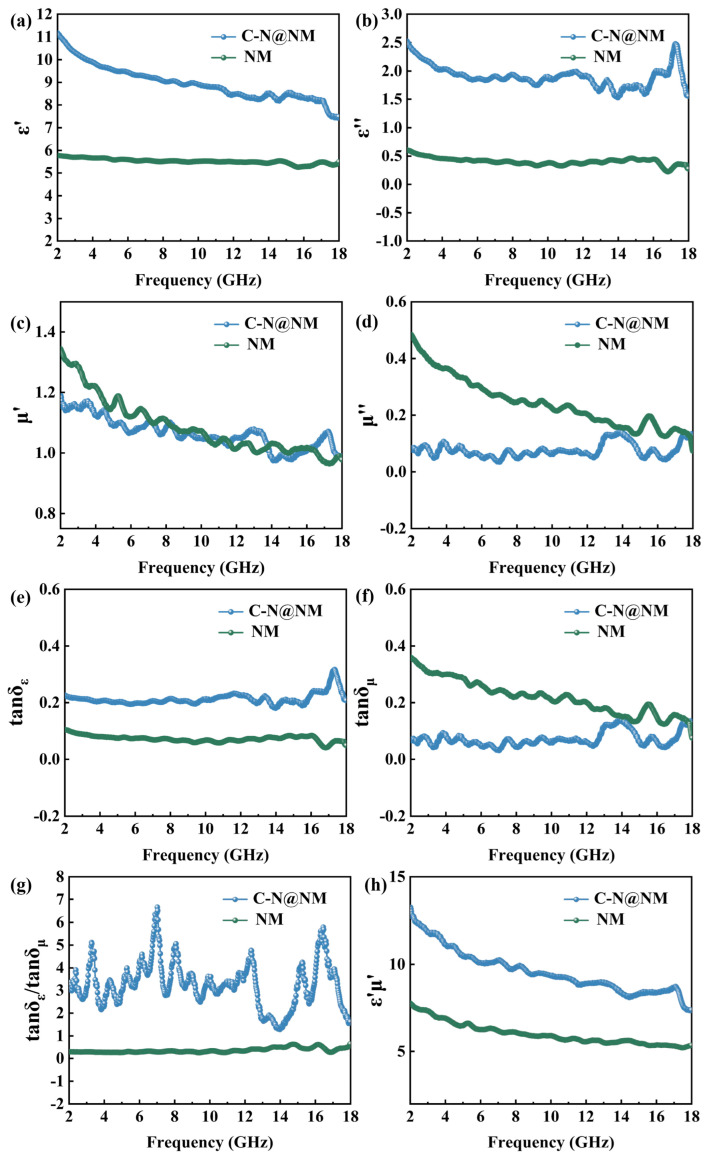
Electromagnetic parameters of NM and C-N@NM: (**a**) $\varepsilon^{\prime }$; (**b**) $\varepsilon^{\prime \prime }$; (**c**) $\mu^{\prime }$; (**d**) $\mu^{\prime \prime }$; (**e**) tan*δε*; (**f**) tan*δμ*; (**g**) tan*δε*/tan*δμ*; (**h**) $\varepsilon^{\prime }\mu^{\prime }$.

**Figure 6 molecules-28-00233-f006:**
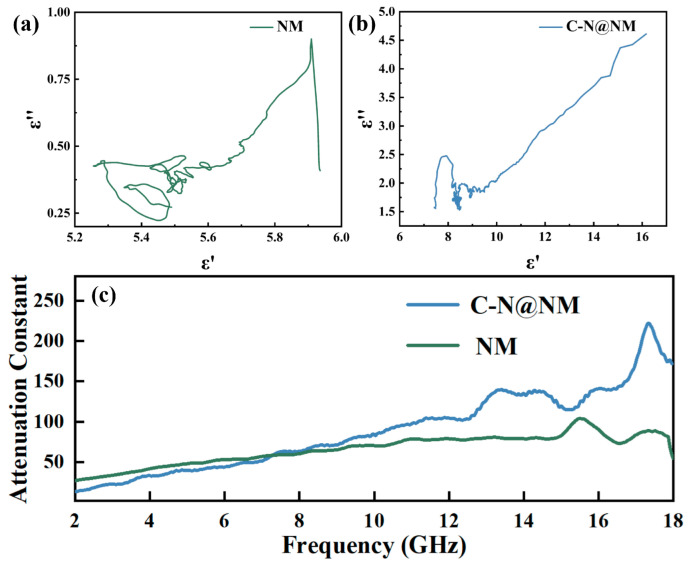
Cole–Cole plots: (**a**) NM; (**b**) C-N@NM; (**c**) Attenuation constants (α) of NM and C-N@NM.

## Data Availability

Data of the compounds are available from the authors.

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
