# Peer review of "In Situ Synthesis of C-N@NiFe2O4@MXene/Ni Nanocomposites for Efficient Electromagnetic Wave Absorption at an Ultralow Thickness Level"

_molecules, 2022, doi:10.3390/molecules28010233_

Round 1

Reviewer 1 Report

This paper “In situ synthesis of C-N@NiFe2O4@MXene/Ni nanocomposites for efficient electromagnetic wave absorption at an ultralow thickness level” presented a synthetic method to obtain the novel composite materials based on Ni/Fe oxide@ MXene as MAMs. Meanwhile, the property of electromagnetic wave absorption was tested and the mechanism was explained as well. The paper is clearly written. The novelty of the results, the scientific style of the presentation, and the deep analysis fully meet the requirements of the Molecules journal. Authors proved that the as-prepared materials are promising as electromagnetic wave absorbers. However, several aspects are recommended to improve and polish before acceptance.

1.     The manuscript shows that the C-N networks structure enhances the dielectric loss of the material. Previous reports have shown that the introduction of carbon can improve dielectric properties. Therefore, the role played by the introduction of N should be demonstrated in the manuscript. In addition, why does the introduction of C-N networks lead to reduction of magnetic loss?

2.     Compared with regular functional materials, the materials presented in this research work is quite complex and require multi-step synthesis. It would be better for readers to fully understand the importance of this research work if the authors describe in detail how to design such complex function materials with excellent properties.

3.     Some key and important research results in absorption field may be mentioned and cited so that we can provide a solid background and progress to the readers, such as Composites Part A, 2018, 115, 371; Nano-Micro Letters, 2022, 14, 173.

4.     Please check the manuscript carefully for spelling and formatting mistakes. For example, in line 118, the “5 ℃-1”.

5.     In figure 1(a), the PDF #89-7218 is seem not in the ICDD-PDF index files, please cheek if it should be PDF #87-0712.

6.     In figure 1(b), there are two PDF #10-0325, one of them should be modified to the PDF card of Ni.

7.     In figure 1, There are duplicate labels in the figure.

8.     In line 165, the authors indicate that N replace the C in sp2 and sp3 C-C bonds. However, XPS provided insufficient evidence. Thus, more convincing evidence should be provided, or the related researches are supposed to be cited.

9.     In figure 2, it would be better if the images of the SEM are presented at the same magnification.

Reviewer 2 Report

In this study, the EM wave absorption properties of NiFe2O4/MXene/Ni were investigated. But before publication below points must be revised.

1. The abstract must be prepared as the : 1. Aim of this paper, 2. brief information materials and methods, and 3. main conclusion

The first sentence of the abstract is not necessary.

2. Define MAMs abbreviation for the first usage.

3. It is necessary to give more information on MXene: please find related information bellow:

Karataş, Y., Çetin, T., Akkuş, İ. N., Akinay, Y., & Gülcan, M. (2022). Rh (0) nanoparticles impregnated on two‐dimensional transition metal carbides,  MXene, as an effective nanocatalyst for ammonia‐borane hydrolysis. International Journal of Energy Research.

4. The transmission line theory (Equations 1,2,3, and 4) given in the introduction must be given in Section 3.

5. Please give information on why NiFe2O4 was chosen for EM wave application.

some information can be found bellow:

Cao, Y., Mohamed, A. M., Mousavi, M., & Akinay, Y. (2021). Poly (pyrrole-co-styrene sulfonate)-encapsulated MWCNT/Fe–Ni alloy/NiFe2O4 nanocomposites for microwave absorption. Materials Chemistry and Physics, 259, 124169.

Akinay, Y., Hayat, F., Kanbur, Y., Gokkaya, H., & Polat, S. (2018). Comparison of microwave absorption properties between BaTiO3/Epoxy and NiFe2O4/Epoxy composites. Polymer Composites, 39(S4), E2143-E2148.

6. There are many studies published on EM wave absorber of Nife2O4 Mxene. Therefore it is important to elaborate on the novelty of this paper. 

7. The synthesis of MXene is not clear. Did you use and etching method? How did you mix Ti3AlC2 and NiCl2? In solution or as a dry powder form?

For example in the sentence of "After Lewis acidic etching, the molten NiCl2 removed the Al atoms of MAX." It was determined that the powders were etched but it was not shown in the materials and method section. 

Round 2

Reviewer 2 Report

The related revisions have been done. I think it is acceptable with its current situation.